# An Improved End-to-End Multi-Target Tracking Method Based on Transformer Self-Attention

Yong Hong [1,2] , Deren Li [1,*], Shupei Luo [3], Xin Chen [3], Yi Yang [3] and Mi Wang [1]

[1] State Key Laboratory of Information Engineering in Surveying, Mapping and Remote Sensing, Wuhan University, Wuhan 430079, China
[2] Mobile Broadcasting and Information Service Industry Innovation Research Institute (Wuhan) Co., Ltd., Wuhan 430068, China
[3] Wuhan Optics Valley Information Technology Co., Ltd., Wuhan 430068, China
* Correspondence: drli@whu.edu.cn; Tel.: +86-13907144816

**Abstract:** Current multi-target multi-camera tracking algorithms demand increased requirements for re-identification accuracy and tracking reliability. This study proposed an improved end-to-end multi-target tracking algorithm that adapts to multi-view multi-scale scenes based on the self-attentive mechanism of the transformer's encoder–decoder structure. A multi-dimensional feature extraction backbone network was combined with a self-built raster semantic map which was stored in the encoder for correlation and generated target position encoding and multi-dimensional feature vectors. The decoder incorporated four methods: spatial clustering and semantic filtering of multi-view targets; dynamic matching of multi-dimensional features; space–time logic-based multi-target tracking, and space–time convergence network (STCN)-based parameter passing. Through the fusion of multiple decoding methods, multi-camera targets were tracked in three dimensions: temporal logic, spatial logic, and feature matching. For the MOT17 dataset, this study's method significantly outperformed the current state-of-the-art method by 2.2% on the multiple object tracking accuracy (MOTA) metric. Furthermore, this study proposed a retrospective mechanism for the first time and adopted a reverse-order processing method to optimize the historical mislabeled targets for improving the identification F1-score (IDF1). For the self-built dataset OVIT-MOT01, the IDF1 improved from 0.948 to 0.967, and the multi-camera tracking accuracy (MCTA) improved from 0.878 to 0.909, which significantly improved the continuous tracking accuracy and reliability.

**Keywords:** transformer; self-attention; multi-view multi-scale; end-to-end; multi-target tracking; raster semantic map; space–time convergence network (STCN)

## 1. Introduction

Vision-based multi-target multi-camera tracking (MTMCT) algorithms are commonly used for trajectory retrieval, action warning, and behavior judgment analysis, and are widely used in frontier fields such as intelligent networked vehicles, vehicle–road collaboration, and satellite image target tracking [1].

The rise of transformer-based multi-target tracking networks [2] has ushered in a new paradigm called "tracking-by-attention" (TBA). Tim M. et al.'s TrackFormer [3] network achieved seamless data association between frames by implementing both position masking and object identity inference through an encoder–decoder self-attention mechanism. Zeng [4] developed a temporal aggregation network for passing temporal correlation query information to aid continuous tracking.

The self-attentive mechanism of the transformer-based multi-target tracking algorithm above is informative, but there are still the following problems in the MTMCT scenario: multiple targets in cross-camera situations, where the target is obscured by the scene or the target moves across the camera, can cause the network to lose continuous tracking of the target. When the target moves within the scene, the changes of position from far to

near, and the rotation, can lead to changes in the scale and viewpoint of the target. All the above situations will affect the re-identification (ReID) accuracy. To solve the above problem, a common approach is to implement the corresponding tracking algorithms separately for different scenarios [5–11]. For example, in the case of overlapping fields of view (FOVs), Berclaz et al. [12] solved the association of targets based on the shortest k-path algorithm, and Hu et al. [13] matched cross-camera trajectories based on pair-wise geometric constraints. In the case of non-overlapping FOVs, Cai et al. [14] matched different targets based on appearance similarity comparison, while Ristani et al. [15] extracted appearance and motion features based on convolutional neural network (CNN) networks to complete target ReID. At the same time, Markov random fields and conditional random fields have been frequently used for target association. For example, Chen et al. [16] used Markov chains and Monte Carlo sampling to obtain the space–time location relationship between cameras and image features for inter-camera association. Chen et al. [17] used Markov random fields to construct an equalized graph model. Chen and Bhanu [18] used conditional random fields to correlate tracklets generated by a single camera. Lee et al. [19] used the inter camera linking model to match temporal, regional, and fusion features.

The above methods are limited to solving target ReID in specific scenes and cannot be applied in complex scenes with multiple views and scales. Thus, the reliability of the methods needs to be improved.

In this study, an end-to-end multi-target tracking method based on transformer self-attention improvement was proposed. Section 2 mainly discusses the basic architecture of the algorithm. In the feature extraction backbone network, YOLOV5 [20] and ResNet50 [21] are fused for multidimensional feature extraction to construct a multidimensional feature library; in the encoder construction, a raster semantic map is associated with multidimensional features to output multidimensional feature vectors with location encoding; in the decoder construction, a space–time convergence network (STCN) is implemented for optimizing the continuous tracking accuracy by conveying contextual information; and in the overall logic processing, a retrospective mechanism based on inverse order processing is added to optimize the overall continuous tracking accuracy. Section 3 is the core of this study, which focuses on the optimization of the decoder in spatial, feature, and logical dimensions by combining the raster semantic map to further improve the multi-target tracking accuracy and reliability. Section 4 describes the data sources, evaluation metrics and detailed experimental procedures. Section 5 presents the optimization results and discussion of this study. Section 6 is the conclusion.

The main contributions of this study are as follows.

A multidimensional feature matching algorithm based on the raster semantic maps was proposed. Based on the walkability information of the raster semantic map, the raster probability matrix was normalized in combination with a Gaussian kernel function to eliminate non-walkable regions and optimize the target localization accuracy. Based on the position encoding input, the multi-camera targets in the overlapping FOVs were clustered to complete the coarse matching of targets across cameras. Based on the multi-dimensional feature library, the targets were matched and tracked with texture features to improve the multi-target tracking accuracy.

A raster semantic map-based spatio-temporal logic matching method was proposed. Using the position encoding input, a weighted inter-projection mechanism based on the Euclidean distance was used to complete the target tracking in the overlapping FOVs. Using the connectivity information of the raster semantic map, the transfer matrix was obtained to calculate the spatio-temporal correlation degree of cross-camera targets, and then combined with the image matching similarity probability. The global graph model was constructed, and the target ID was obtained using the minimum flow solution, thus realizing cross-camera non-overlapping FOVs region target tracking and improving the reliability of multi-target tracking.

## 2. Transformer-Based End-to-End MTMCT Algorithm Architecture

In this study, an end-to-end multi-objective tracking method was proposed based on the transformer's self-attentive [22] improvement. The proposed approach continuously tracked pedestrians in complex situations (e.g., cross-camera, multi-view, and multi-scale) and constructed a raster semantic map to encode target locations. Cross-camera targets were continuously tracked based on three dimensions (i.e., temporal, spatial and logical), and a STCN was constructed to transfer relevant feature parameters. A retrospective mechanism was also added to optimize the historical target tracking results by inverting the order and normalizing the "overlapping field of view" and "non-overlapping field of view" scenarios in cross-camera tracking as shown in Figure 1.

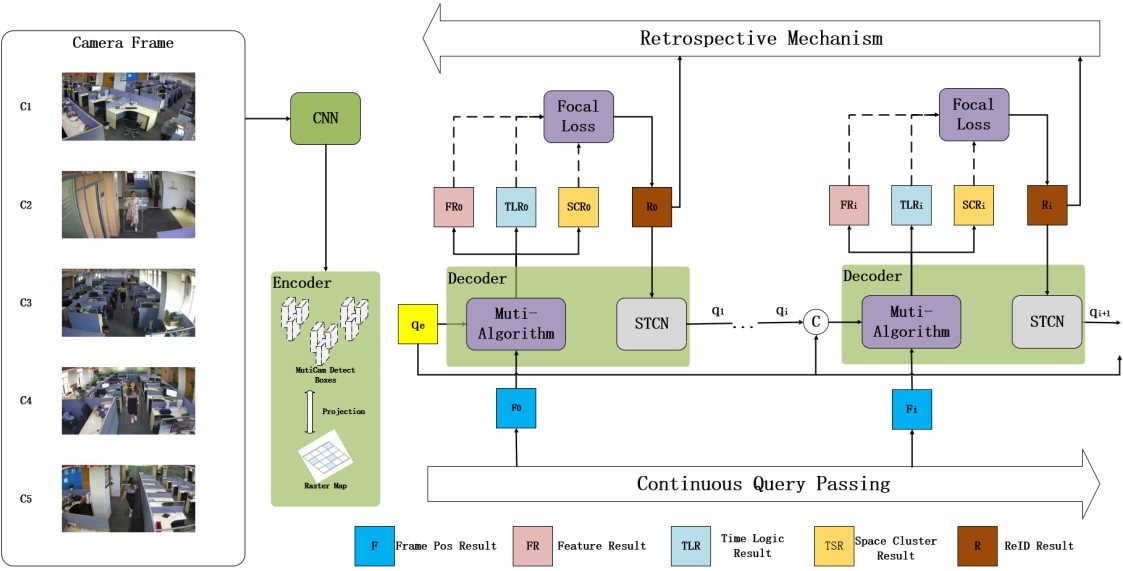

**Figure 1.** Overall algorithm structure.

The steps are as follows:

(a) For the multi-view camera detection, the corresponding detection frames and texture features were first obtained based on a multi-dimensional feature extraction CNN network and fed into the encoder;

(b) The encoder received the raster semantic map, which was constructed based on the target scene. Using the projection of the multi-dimensional feature detection frame and the raster semantic map from the multi-view detection results, the final detection frame result (Frame Pos Result) in the object space was obtained and sent into the decoder;

(c) The decoder received the frame detection result from the encoder and the a priori query of the previous frame. The decoder consisted of three parts: the spatial clustering and semantic filtering algorithm that generated the spatial clustering results, the multi-dimensional feature dynamic matching algorithm combined with the raster semantic map filter that produced the feature, and the space–time logic-based multi-visual target tracking algorithm that created the logic result. The results were subjected to focal loss to obtain the continuous tracking ReID result of the current frame, which was input to the STCN;

(d) The STCN produced an a priori query for the next frame and cascaded it with the historical query. It was then fed it into the decoder and repeated (a) (b) (c) in the algorithm for the next frame;

(e) When the overall tracking had been completed, the ReID was optimized by reviewing the overall results using inverted order processing and compensating for the confidence score in the historical results.

### 2.1. Construction of Backbone Network and Encoder

In the proposed method, YOLOV5 [20] and ResNet50 [21] were fused to construct a multi-dimensional feature extraction backbone network based on the pre-trained convolutional neural network YOLOV5. Pedestrian and head detection frames were extracted using regression to obtain candidate target locations, while multi-target texture features based on ResNet50 were obtained to produce multi-dimensional feature vectors [23].

In constructing the transformer-based encoder, the raster vector was associated with the camera field of view based on pre-constructed raster semantic map data to obtain the position encoding of the target in a particular scene and fuse it with a multi-dimensional feature vector. The encoder outputs were the target position encoding and the multi-dimensional feature vector.

### 2.2. Construction of a Transformer-Based Decoder

The decoder was divided into three parts: location encoding output based on the raster semantic map, multi-algorithm, and STCN. The multi-algorithm part refers to the idea of a multitask learning network [24], and the multidimensional feature matching algorithm and the space–time logic matching algorithm were constructed based on the location encoding provided by the raster semantic map to use the location encoding in the target re-identification task as complementary information [24]. The overall structure is shown in Figure 2 below. The multi-algorithm of the decoder is described in detail in Section 3.

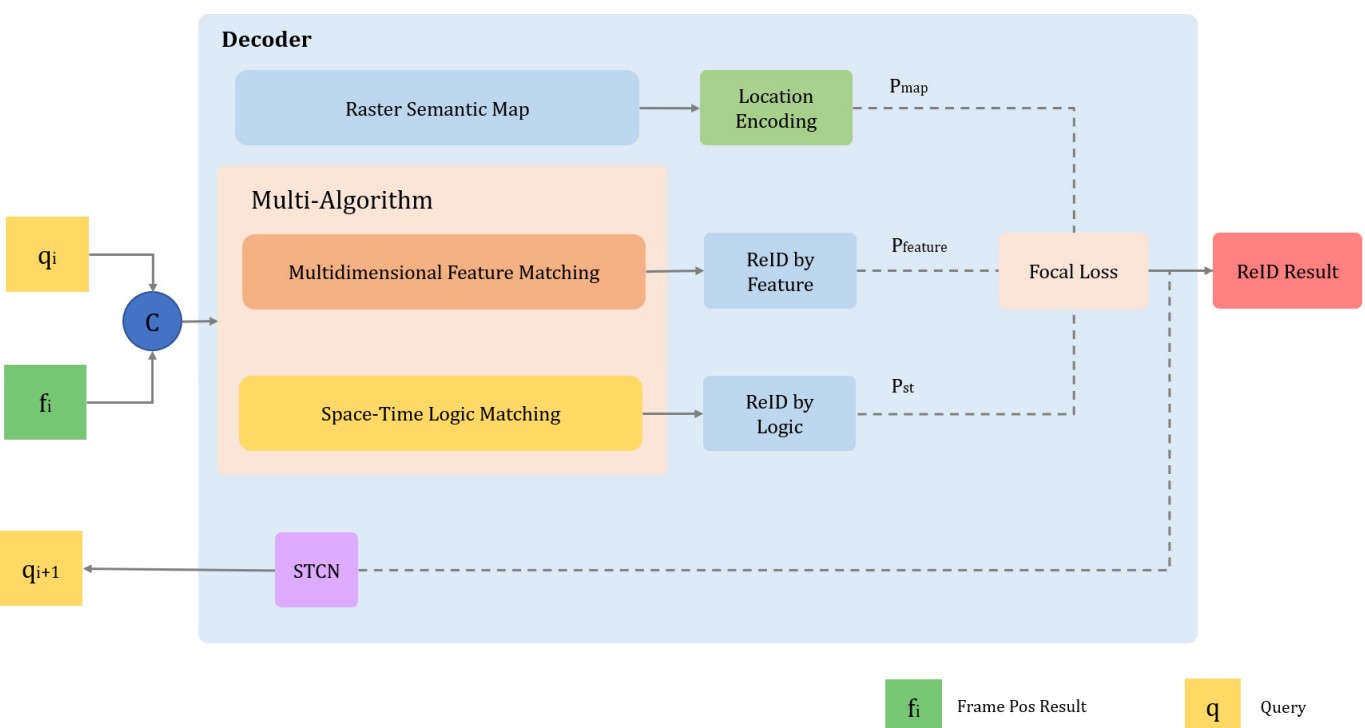

**Figure 2.** Decoder structure.

The ReID problem was resolved by iterative delivery of tracking ensemble queries, while the generation and delivery of tracking ensemble queries required consideration of the space–time correlation, image continuity, and other issues. In this study, a space–time convergence network was constructed with enhanced temporal correlation to provide contextual a priori information for continuous target tracking, as shown in Figure 3 below.

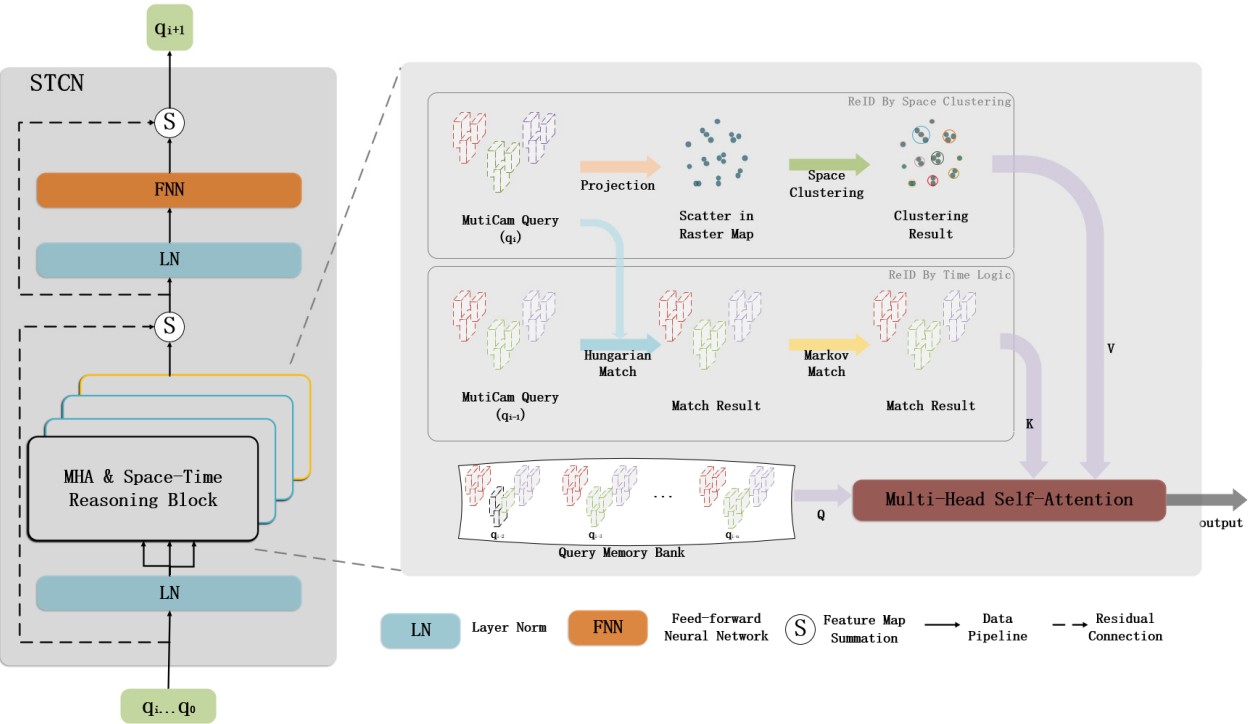

**Figure 3.** STCN structure diagram.

The STCN involves the following steps:

(a) Spatial clustering based on a multi-camera query was used to obtain query input under spatial logic;

(b) Hungarian matching algorithm and Markov random field matching based on the query of consecutive frames were incorporated to obtain the query input under temporal logic;

(c) Based on inputs (1) and (2), a query ensemble $q_{bank}$ was constructed in combination with a historical continuous frame tracking query, while a cascade of multiple queries was implemented as follows:

$$q_{bank} = \left\{ \widetilde{q}_c^{\,i-M}, \dots, \widetilde{q}_c^{\,i} \right\}$$
$$tgt = \widetilde{q}_c^{\,i-M} \oplus \cdots \widetilde{q}_c^{\,i-1} \oplus \widetilde{q}_c^{\,i} \tag{1}$$

where $tgt$ denotes a cascade of queries.

(d) The cascaded queries were fed into the multi-attention module to generate attention weights, resulting in the following dot product attention formula.

$$q_{sa}^i = \sigma_s \left( \frac{tgt \cdot tgt^T}{\sqrt{d}} \right) \cdot \bar{q}_c^i \tag{2}$$

where $\bar{q}_c^i$ is taken as the multiple attention (MHA) query, $\sigma_s$ denotes the softmax function, and $d$ indicates the dimension of the track query.

(e) Further tuning and optimization based on the feed-forward network (FFNN) are employed to finally output the track query $q_t^{i+1}$ for the next frame.

$$t\widetilde{g}t = LN\left( q_{sa}^i + \bar{q}_c^i \right)$$
$$\hat{q}_c^i = LN(FC(\sigma_r(FC(\widetilde{g}\widetilde{g}t)) + t\widetilde{g}t)) \tag{3}$$

where $FC$ denotes a linear projection layer, and $LN$ denotes layer normalization.

(f) In the next frame detection, $q_t^{i+1}$ is used as the input of the decoder, combined with the encoded output of the image, to complete the subsequent target tracking.

### 2.3. Construction of a Retrospective Mechanism Based on Inverse Order Processing

In this study, there are multiple dimensions of the target re-identification mechanism. In the positive-order real-time processing, there is a situation where the continuous tracking target IDs are only optimally matched in the middle section of the tracklet, in which case, there are unmatched IDs in the historical tracklet, thus affecting the overall accuracy.

Therefore, a retrospective mechanism was constructed (Figure 4) to process the relevant tracklet in reverse order after completing the overall tracking and optimizing the historical target IDs.

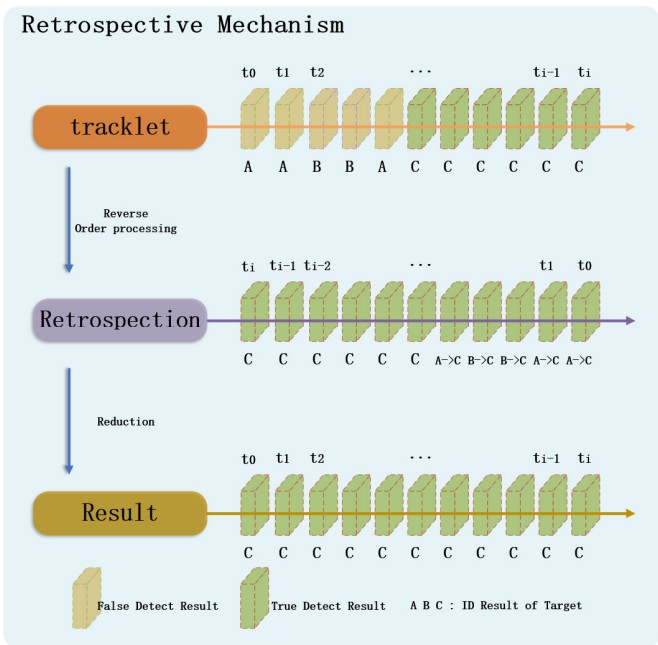

**Figure 4.** Schematic diagram of tracklet-based retrospective processing.

### 2.4. Collective Average Loss

In this study, we constructed a self-attentive model based on STCN for spatial-temporal modelling of multi-target tracking quer, and performed a loss calculation based on continuous multi-frame tracking prediction results with the following loss function [4]:

$$
\begin{gathered}
CAL = \frac{\sum L_i}{\sum V_i} \\
L_i = \lambda_{cls}\, L_{cls} + \lambda_{L_1}\, L_{L_1} + \lambda_{iou} L_{iou} = \lambda_{cls}\, (\, log(P_{feature} * P_{st} * P_{map})) + \\
\lambda_{L_1} \left\| b_i - \acute{b_i} \right\| + \lambda_{iou}\, C_{iou}(b_i,\, \acute{b_i})
\end{gathered}
\tag{4}
$$

where $CAL$ denotes collective average loss, $V_i$ is the total number of targets, $\lambda_{cls}, \lambda_{L_1}, \lambda_{iou}$ are the weight parameters, $P_{feature}, P_{st}, P_{map}$ are the feature matching probability (Figure 2), space–time matching probability and semantic map matching probability respectively, $b_i$ is the current detection frame, $\acute{b_i}$ is the real detection frame, and $C_{iou}$ is the result of the IoU(intersection over union) loss function calculation.

## 3. Optimization of the Decoder Based on the Raster Semantic Map

### 3.1. Multidimensional Feature Matching on the Raster Semantic Maps

By introducing the walkability semantic information of the raster semantic map, our method normalized the probability of walkability to filter the target data and eliminate gross errors. At the same time, the co-visibility semantic information was used to constrain the location results for improving the accuracy of the target positioning. The construction method and main process of the raster semantic map are discussed in the first part of Section 4.

According to the calibrated camera's intrinsic and extrinsic, the detection pixel coordinates of the target are mapped to the raster map C, and the actual object coordinates of the target were calculated. The camera projection model is:

$$
\begin{bmatrix} u \\ v \\ 1 \end{bmatrix} = \begin{bmatrix} f & 0 & u_0 & 0 \\ 0 & f & v_0 & 0 \\ 0 & 0 & 1 & 0 \end{bmatrix} \begin{bmatrix} R & t \\ 0^T & 1 \end{bmatrix} \begin{bmatrix} X \\ Y \\ Z \\ 1 \end{bmatrix}
\tag{5}
$$

where $X$, $Y$ and $Z$ are the three-dimensional coordinates, $u$ and $v$ are the pixel coordinates of target detection, $f$ is the focal length of the camera, $u_0$ and $v_0$ are the coordinates of the principal point.

Since the height of the camera from the ground is known and $Z$ is a fixed constant, coordinates $X$ and $Y$ on the ground plane can be solved through the pixel coordinates of the target $(u,v)$, and the corresponding initial coordinates $C^0$ of the current target on the raster map can be obtained by querying the raster map.

$$
C^0 = (l, m)
\tag{6}
$$

Select the target location and 8 surrounding points on the raster map to form a $3 \times 3$ candidate region. As shown in Figure 5, by querying the walkability semantic information, the walkability matrix of the candidate region is obtained. Take the discrete Gaussian distribution model:

$$
g(i,j) = \frac{1}{2\pi\sigma^2} e^{-\frac{(i^2+j^2)}{2\sigma^2}}, \forall\, i,j \in \{-1,0,1\}
\tag{7}
$$

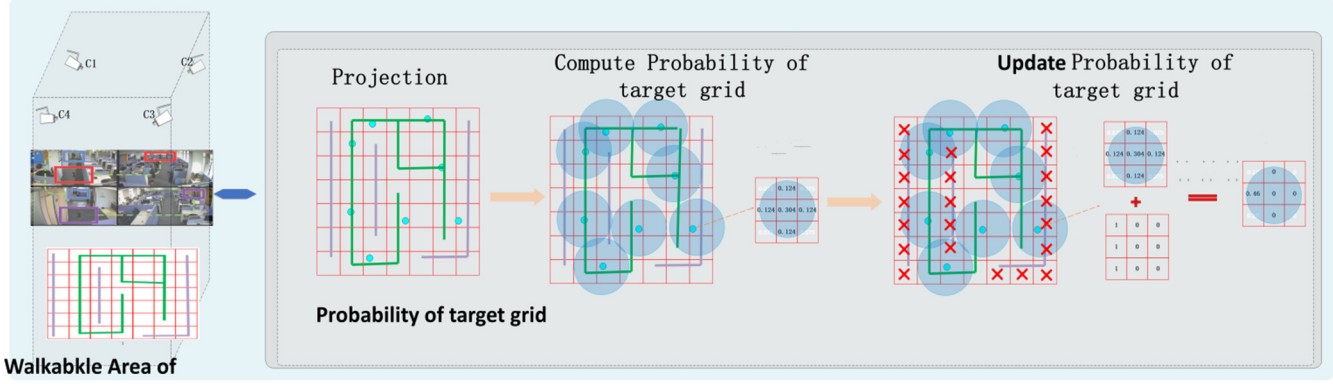

**Figure 5.** Walkability semantic information.

The target probability grid matrix is built.

$$
P_o = \begin{bmatrix} 0.075 & 0.124 & 0.075 \\ 0.124 & 0.304 & 0.124 \\ 0.075 & 0.124 & 0.075 \end{bmatrix}
\tag{8}
$$

The candidate region probability matrix $P_1$ is obtained by multiplying the elements in the matrix $P_c$ by the corresponding elements in the matrix $P_0$.

$$
P_1(i,j) = P_0(i,j) * P_c(i,j), \forall\, i,j \in \{0,1,2\}
\tag{9}
$$

The normalized candidate region probability matrix $P_2$ is obtained by normalizing $P_1$.

$$
P_2 = \frac{P_1}{|P_1|}
\tag{10}
$$

The index of the maximum value of $P_2$ indicates the relative position of the maximum probability in the candidate region. According to the grid coordinates, the precise positioning coordinates $C^1$ can be determined.

$$\left(\widetilde{i}, \widetilde{j}\right) = argmax\ (P_2(i,j)),\ \forall\ i,j \in \{0,1,2\}$$
$$C^1 = (l+1-i, m+1-j) \tag{11}$$

Thus, accurate location information can be obtained under the constraint of the raster semantic map.

After the coordinates were precisely located, the camera co-visibility semantic information was used to map the targets from multiple cameras to the raster semantic map in the overlapping FOV region, and spatial clustering was conducted according to the distance of the grid where they were located. The closed targets were matched as the same ones. Employing NTP time synchronization, the sampling time reference of multiple cameras can be guaranteed to be consistent, and the Euclidean distance between each target point can be calculated frame by frame.

$$D(O_1, O_2) = |C^{o_1} - C^{o_2}| \tag{12}$$

We set the positioning accuracy error of the same target under different cameras to be within one grid distance. At the initial matching, the targets within distance $D(O_1, O_2) \leq 1$ are marked as the same one. Then the texture feature vector $w_1$ is extracted from these targets. With the global feature library texture feature vector $w_2$, the similarity $L(w_1, w_2)$ can be calculated in cosine distance.

$$L(w_1, w_2) = \frac{w_1 w_2}{|w_1||w_2|} \tag{13}$$

The one with the largest similarity is the best match. If $L \geq 0.75$ the matching target ID will be the output. If $L < 0.75$, the ID and associated features will be added to the global feature library, and the feature matrix of the global feature library will be updated at the same time, as shown in Figure 6.

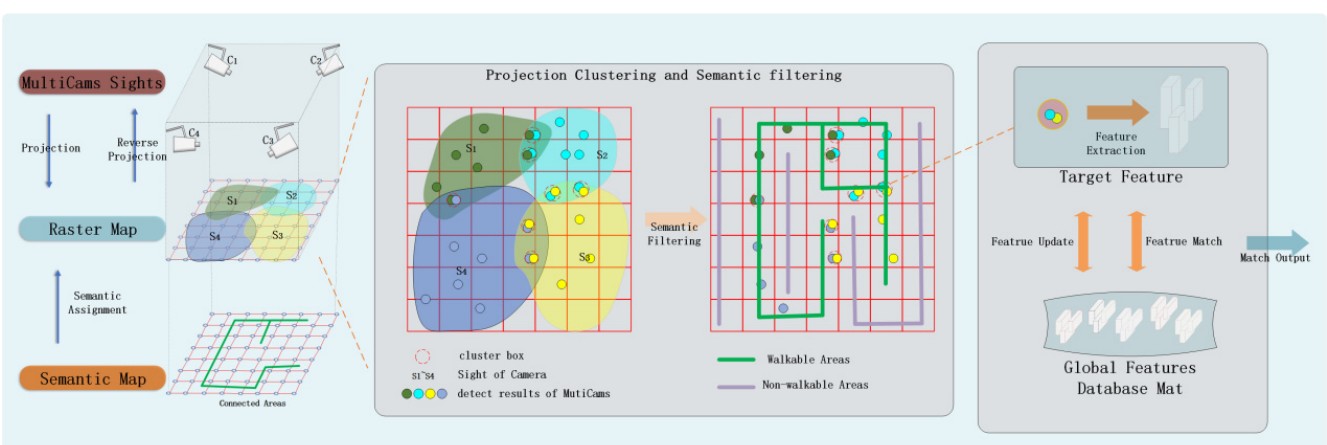

**Figure 6.** Co-visibility semantic information and coarse matching.

The multi dimension feature matching method combined with the raster semantic map introduced the global feature library, which solved the ID uniqueness problem between multi-cameras and multi-objects. The co-visibility semantic information of raster semantic map was used to reduce the feature mismatch caused by multi-cameras and multi-scale.

### 3.2. Space-Time Logic Matching Based on the Raster Semantic Maps

In the case of a single camera, based on the continuous frame position encoding input, the bipartite graph optimal matching operator [25] is used as the core to achieve complete matching of front and back frames and obtain the target tracking results of continuous detection frames in a single camera field of view to achieve uniformity of ID and confidence.

For cross-camera overlapping field of view scenes, a weighted interjection mechanism was developed to calculate the Euclidean distance between multi-target detection results between cameras based on cross-camera position coding input and set a dynamic threshold upper limit. When the Euclidean distance between the two inter-camera results is the smallest and the distance is less than the upper dynamic threshold, the two targets are considered aligned. The aligned detection results will share the same ID and confidence score, which means the one with higher confidence score will overwrite the other one. Hence, alignment of the multi-camera view tracking trajectory is achieved.

For the cross-camera overlapping FOVs scenes, the global Markov random field [16] was constructed based on the location-encoded input with the camera as the node to calculate the transfer probability of the current target in the raster semantic map, as shown in Figure 7. An n-step transfer matrix $P(N)$ can be obtained, and the probability of a target located at camera j reaching camera j is $P_{ij}$.

$$P(N) = \begin{bmatrix} P_{11}(N) & P_{12}(N) & \cdots & P_{1n}(N) \\ P_{21}(N) & P_{22}(N) & \cdots & P_{2n}(N) \\ \cdots & \cdots & \cdots & \cdots \\ P_{n1}(N) & P_{n2}(N) & \cdots & P_{nn}(N) \end{bmatrix} \tag{14}$$

$$P_{ij} = max P_{ij}(N) \tag{15}$$

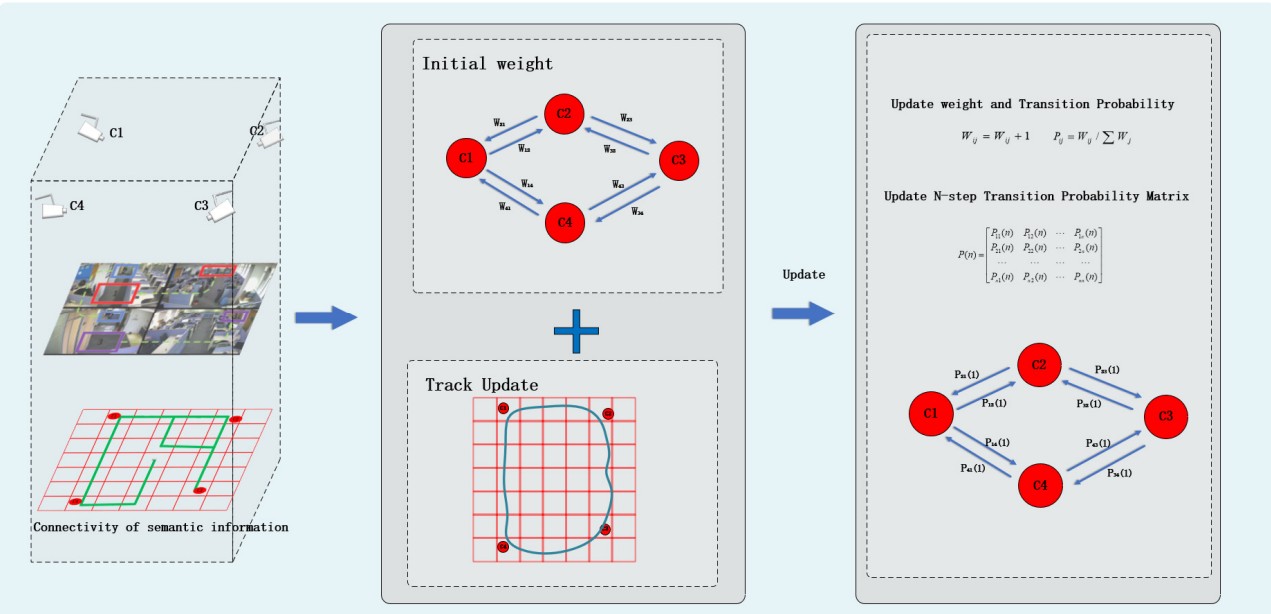

**Figure 7.** Semantic information on connectivity.

When the target disappears from the camera field of view, the transfer matrix can be used to predict which field of view the target will reappear.

St-Reid was applied to extract the tracklet image information and space–time information. Given tracklets $L_i$, $L_j$ and their feature vectors $x_i$, $x_j$, $P_a$ denote the probability of a similarity of image information between $L_i$ and $L_j$. $P_t$ denote the probability of space–time association between $L_i$ and $L_j$ that is, whether the target in the last frame of $L_i$ and the target

in the first frame of $L_j$ belongs to the same object. So, the probability density function of $P_t$ is related to camera ID, target ID and time. $P_a$ and $P_t$ are given by the following equations:

$$P_a(L_i, L_j) = \frac{x_i x_j}{|x_i||x_j|} \tag{16}$$

$$P_t(L_i, L_j) = P(m_{i1} = m_{i2}|c_{j1}, c_{j2}, t_{k2} - t_{k1}) \tag{17}$$

In which $m_{i1}$, $m_{i2}$ are the target ID, $c_{j1}$, $c_{j2}$ are the camera ID, $t_{k2}$ is the time the target $m_{i1}$ leaves the camera $c_{j1}$, and $t_{k2}$ is the time the target $m_{i2}$ appears in the camera $c_{j2}$.

The Markov random field [12] was used as the base model to construct the global graph with the start and end of the tracklet as nodes. The definitions and weights of the edges of the global graph(see Figure 8) are as follows.

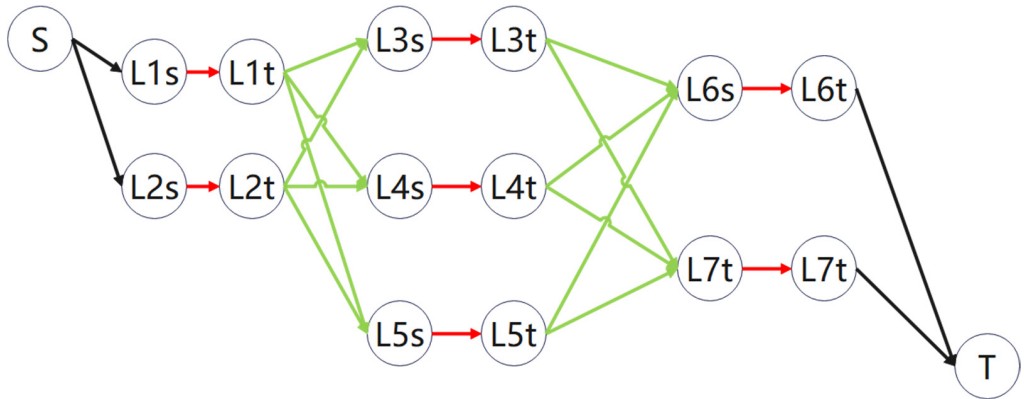

**Figure 8.** Global graph model (EGM), 3 steps, 7 tracklets.

(a) The red edges indicate the connection between the start moment $t_i^e$ and the end moment $t_i^s$ of the tracklet. The probabilities and weights are computed by the following expressions:

$$c_i = P(L_i|\Gamma) = \frac{\sum_{k=t_i^s}^{t_i^e} \alpha_k}{t_i^e - t_i^s} \tag{18}$$

$$w(L_i) = -log\frac{c_i}{1 - c_i} \tag{19}$$

where $L_i$ denotes the tracklet, $\Gamma$ is the set of trajectories, $\alpha_k$ is the probability of similarity between the frames of the tracklet, $c_i$ is the probability that the tracklet $L_i$ holds, and $w(L_i)$ is the probability that the tracklet $L_i$ is the weight of the tracklet in the graph.

(b) The green edges indicate connections between tracklets given by the following weights.

$$w(Li|Lj) = -k_a \, logP_a(L_iL_j) - k_t \, log \, P_t(L_iL_j) \tag{20}$$

(c) The black edges indicate edges connected at the start and end nodes with a weight of zero.

The min-cost flow method [26] is used to obtain the relationships between tracklets and the corresponding IDs. The optimal set of tracks can be calculated using the following:

$$\Gamma^* = arg \, max_\Gamma \prod_i P(L_i|\Gamma) \prod_{\Gamma_k \in \Gamma} P(\Gamma_k)$$

$$= arg \, max_\Gamma \prod_i P(L_i|\Gamma) \prod_{\Gamma_k \in \Gamma} \prod_{L_{k_1}, L_{k_2,...} \in \Gamma_k} P(L_{k_{j+1}}|L_{k_j}) \tag{21}$$

$$\Gamma_i \cap \Gamma_j = \Phi, \, \forall \, i \neq j$$

Using this approach, the tracking association for multiple tracklets in a cross-camera non-overlapping field-of-view scenario can be determined, enabling the transfer of IDs.

## 4. Experiments

### 4.1. Materials

#### 4.1.1. Image Data

The MOT17 [27] public dataset was used in this study, along with the self-built loop-tracking dataset OVIT-MOT01.

MOT17 is a standard dataset proposed in 2017 for measuring multi-target detection and tracking methods.

The self-built loop tracking dataset OVIT-MOT01 was constructed from video captured by five cameras, arranged in a zigzag office area, and calibrated for intrinsic and extrinsic parameters. It contains 10,105 consecutive images and 8299 detection frames to evaluate the accuracy of cross-camera pedestrian re-identification and tracking.

#### 4.1.2. Raster Semantic Map Data Construction

Figure 9 presents the general procedure for generating the semantic raster maps [21–23]. The main steps are as follows:

(a) Joint calibration and area association stitching based on multiple view cameras in the scene were used to obtain a location association map in the corresponding pixel space;

(b) The global image based on the camera pixel space resolution was gridded to obtain a raster vector base map. A pointer matrix $C_{ij}$ based on the raster with coordinates (i, j) was then constructed to represent the raster attributes;

(c) Semantic information was sequentially generated on the co-visibility, walkability, and connectivity of the raster semantic map [28–30];

(d) Semantic information on the co-visibility of raster maps based on camera calibration parameters, the projection of the camera field of view into object space S1 to S4, and information on the currently visible raster were recorded;

(e) Semantic information on the walkability and connectivity of the raster map was projected onto the raster map based on the base map or motion trajectory. The walkability and connectivity information of the current raster was then recorded.

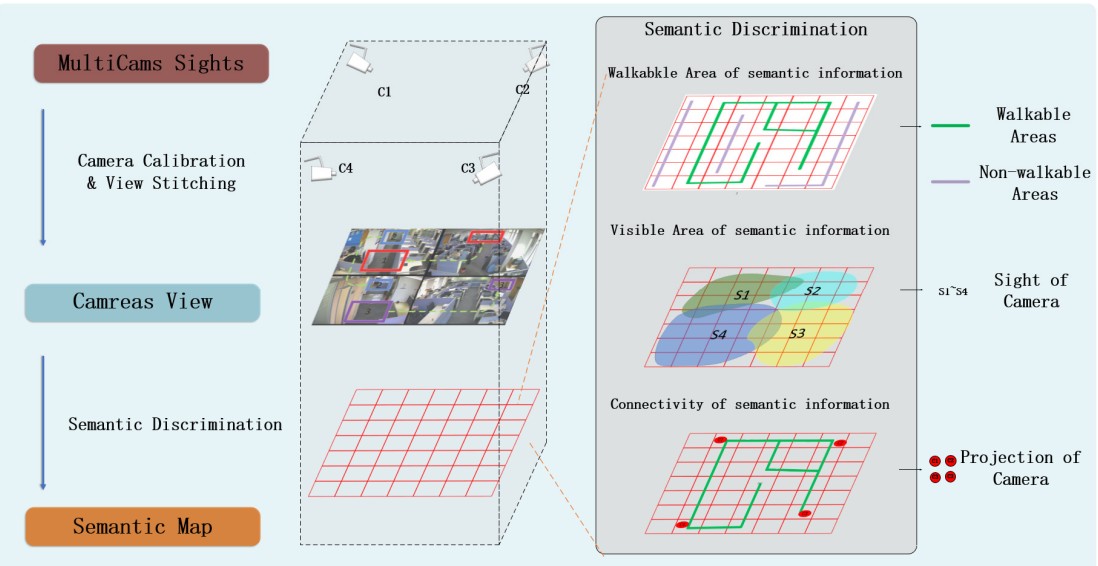

**Figure 9.** Semantic raster map construction.

### 4.2. Evaluation Metrics

The Identification F1-score (IDF1) [31,32], the multiple object tracking accuracy (MOTA) [33–36], and the multi-camera tracking accuracy (MCTA) [37–40] were used as the outcome criteria in this study:

(a) IDF1, the summed average of Identification Precision (IDP) and Identification Recall Rate (IDR), was used to assess the degree of identification accuracy;

(b) MOTA was used to assess the reliability of target tracking;

(c) MCTA, a multi-camera tracking metric, was used to assess the tracking accuracy of multiple cameras.

### 4.3. Implementation Details

The above method was implemented and evaluated on MOT17 and OVIT-MOT01. MOT17 is a public dataset used for accuracy validation and cross-sectional comparison with other algorithms. OVIT-MOT01 is a self-built dataset for validating single and multi-camera target tracking accuracy, comparing the annotation results, and generating the accuracy reports. Moreover, an ablation study was presented to verify the contribution of each component.

### 4.4. Validation of Single Camera Accuracy Results Based on the Publicly Available Dataset (MOT17)

Accuracy validation was performed using the MOT17 dataset, and the results are summarized in Table 1. According to Table 2, the IDF1 value of this research method reaches 0.782 and the MOTA value 0.836 in the public dataset MOT17. Compared with the public algorithms in the current MOT17 ranking, the MOTA accuracy of this study was ranked first and the IDF1 value was ranked fourth, which shows that the target tracking algorithm of this research can reach the average level of the current leading algorithms.

**Table 1.** MOTA values based on MOT17.

|  | IDF1 | MOTA | IDP | IDR | Recall | Precision |
|---|---|---|---|---|---|---|
| **MOT17-02-SDP** | 0.577433 | 0.666111 | 0.682192 | 0.500565 | 0.707013 | 0.963547 |
| **MOT17-04-SDP** | 0.907841 | 0.945097 | 0.919049 | 0.896903 | 0.961099 | 0.984831 |
| **MOT17-05-SDP** | 0.735971 | 0.788926 | 0.809045 | 0.675004 | 0.816684 | 0.97886 |
| **MOT17-09-SDP** | 0.643427 | 0.782535 | 0.696413 | 0.597934 | 0.824601 | 0.960411 |
| **MOT17-10-SDP** | 0.648384 | 0.730119 | 0.716602 | 0.592024 | 0.784952 | 0.950127 |
| **MOT17-11-SDP** | 0.835397 | 0.873145 | 0.860902 | 0.811361 | 0.910237 | 0.965816 |
| **MOT17-13-SDP** | 0.727051 | 0.801151 | 0.767242 | 0.690861 | 0.855437 | 0.950014 |
| **OVERALL** | 0.781575 | 0.83606 | 0.828008 | 0.740073 | 0.868322 | 0.971496 |

**Table 2.** Comparison of open algorithm metrics based on MOT17.

| Method | IDF1 | MOTA |
|---|---|---|
| SelfAT [31] | 0.798 | 0.800 |
| ByteTrack [32] | 0.773 | 0.803 |
| QuoVadis [33] | 0.777 | 0.803 |
| FOR_Tracking [34] | 0.777 | 0.804 |
| BoT_SORT [35] | **0.802** | 0.805 |
| BYTEv2 [36] | 0.789 | 0.806 |
| MiniTrackV2 [41] | 0.788 | 0.818 |
| Ours | 0.782 | **0.836** |

The results of the proposed method were compared with other published algorithms, and the comparative summary is presented in Table 2.

### 4.5. Continuous Tracking Accuracy Based on the Self-Built Dataset OVIT-MOT01

Figure 10 shows the continuous tracking accuracy results in single-camera scenes using the OVIT-MOT01 dataset.

## MOTA VALUES

**■ Transformer ■ Feature ■ Logic**

| | Cam1 | Cam2 | Cam3 | Cam4 | Cam5 | Total |
|---|---|---|---|---|---|---|
| ■ Transformer | 0.898 | 0.757 | 0.904 | 0.729 | 0.634 | 0.819 |
| ■ Feature | 0.898 | 0.757 | 0.905 | 0.729 | 0.772 | 0.853 |
| ■ Logic | 0.91 | 0.772 | 0.914 | 0.743 | 0.76 | 0.86 |

**Figure 10.** OVIT-MOT01 MOTA values for each camera. Note: The MOTA values for each of the five cameras were tested in three cases (transformer temporal logic tracking; transformer + multi-dimensional feature dynamic matching; transformer + multi-dimensional feature dynamic matching + temporal logic matching).

### 4.6. Ablation Experiments Based on OVIT-MOT01

The overall cross-camera re-recognition ablation experiments of the five cameras were tested in four scenarios (transformer temporal logic tracking; transformer + multi-dimensional feature dynamic matching; transformer + multi-dimensional feature dynamic matching + temporal logic matching; transformer + multi-dimensional feature dynamic matching + temporal logic matching + retrospective mechanism); the summary of results is presented in Figure 11.

## RESULTS OF REID

**■ Transformer ■ Feature ■ Logic ■ Retrospective**

| | IDF1 | MOTA | MCTA |
|---|---|---|---|
| ■ Transformer | 0.417 | 0.819 | 0 |
| ■ Feature | 0.933 | 0.853 | 0.86 |
| ■ Logic | 0.948 | 0.86 | 0.878 |
| ■ Retrospective | 0.981 | 0.863 | 0.909 |

**Figure 11.** Results of ReID ablation experiments based on OVIT-MOT01.

## 5. Discussion

The MOTA for the single-camera target tracking using the MOT17 dataset reached 0.782, while the IDF1 value was 0.836 (see Tables 1 and 2) which significantly outperforms the current state-of-the-art method MiniTrackV2 [41] by 2.2%. Since the focus of this study is on complex scenarios with multiple cameras and scales, the subsequent analyses were based on the OVIT-MOT01 dataset. Using the transformer-based tracking matching, the overall MOTA for the OVIT-MOT01 dataset reached 0.819 while the IDF1 value was only 0.417 (see Figures 10 and 11). The image accuracy results were affected by the following factors: (1) ID switching due to masking; (2) ID switching due to the target entering and leaving the camera; and (3) target ID switching due to perspective and scale shifts, as detailed in Figure 12a–c.

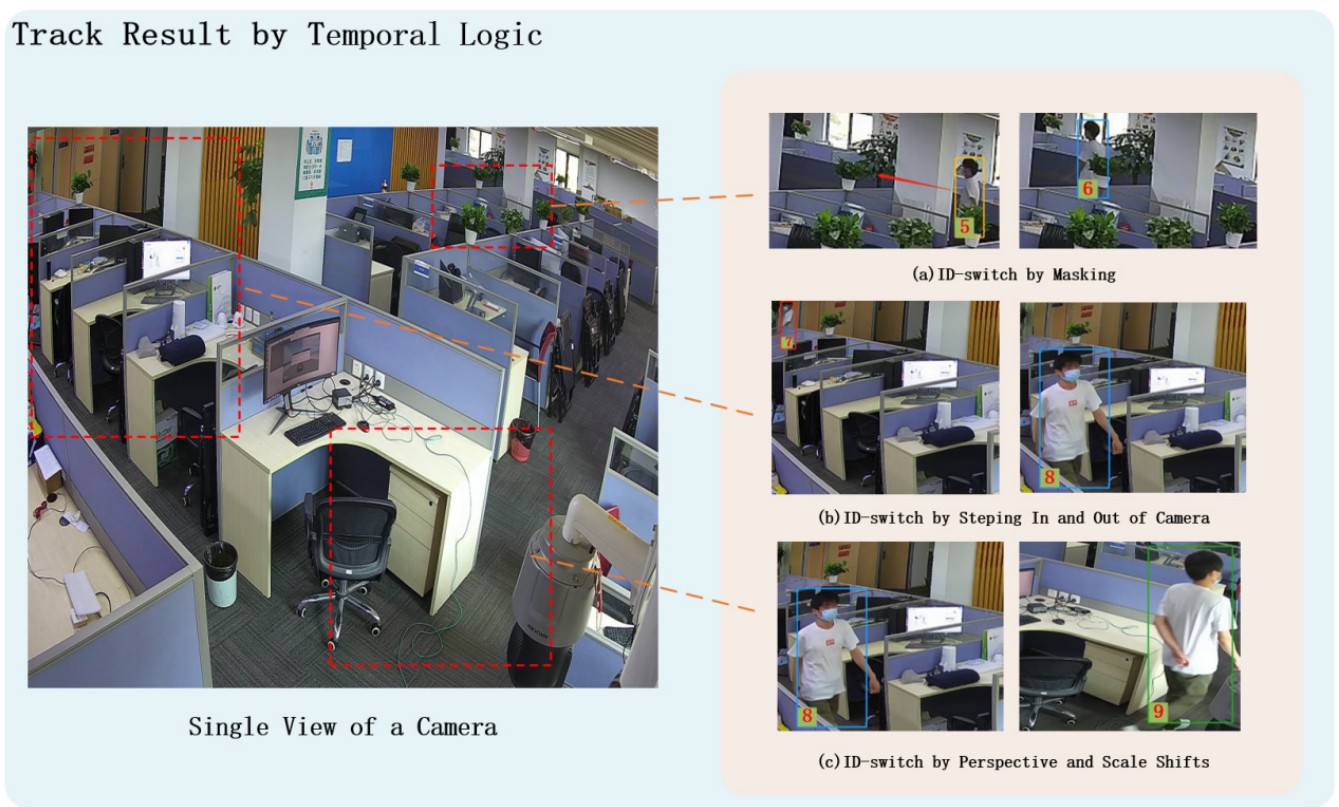

**Figure 12.** Graph of continuous tracking results based on the transformer's temporal logic.

### 5.1. Optimization Results Based on Multi-Dimensional Dynamic Feature Matching Method

After the multi-dimensional dynamic feature matching, the overall MOTA using the OVIT-MOT01 dataset improved to 0.853, the MCTA reached 0.860, and the IDF1 significantly increased to 0.933 (see Figure 11). The feature-based matching is independent of time and space and can provide accurate re-identification of targets. The constructed multi-dimensional feature library effectively compensates for the problem of false detection caused by changes in illumination, angle, and scale. After matching, targets lost due to occlusion can be re-tracked after re-emergence. The ID switch situation generated by targets when crossing cameras is significantly reduced, and the ID swap due to target interleaving is eliminated, resolving the target loss problem caused by viewpoint changes. As shown in Figure 13, the ID switches due to masking, cross-camera, perspective shifts, and scale shifts in Figure 12 have been optimized.

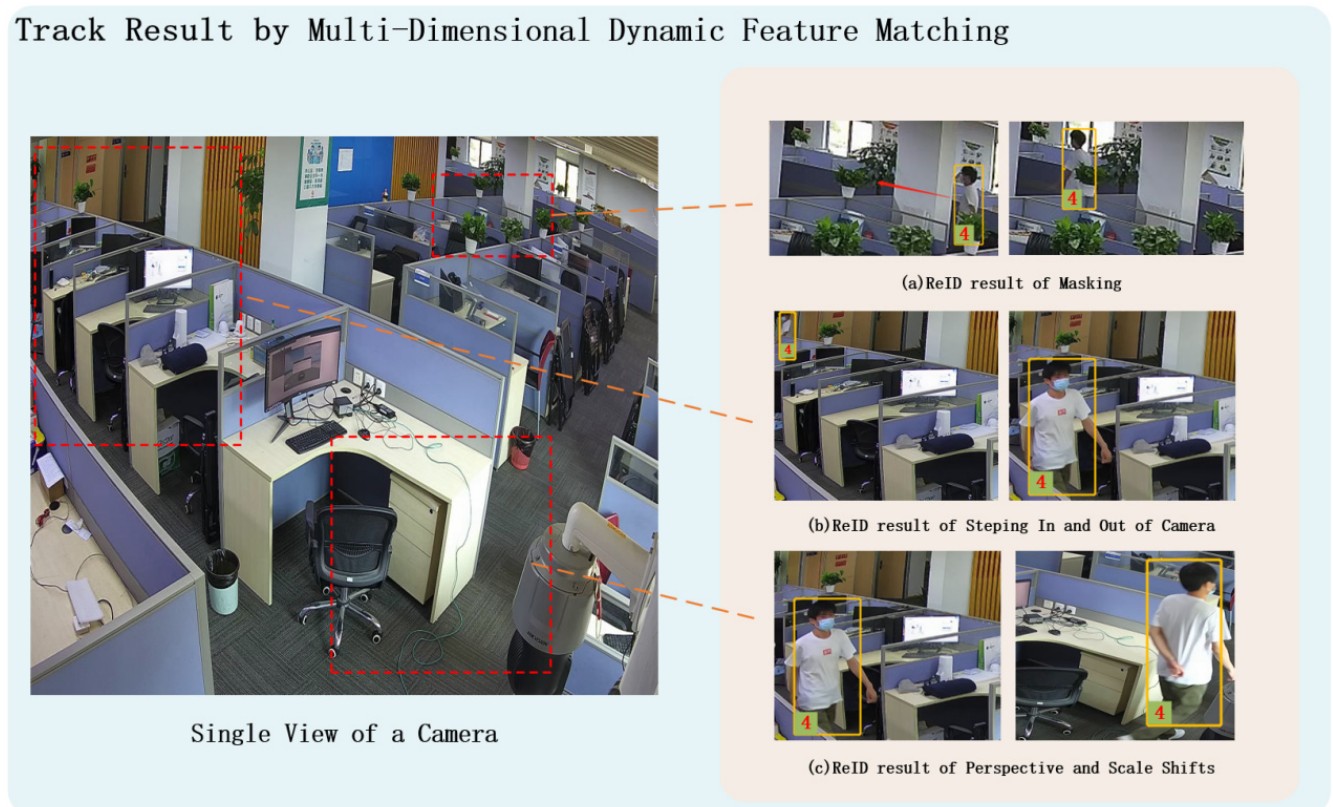

**Figure 13.** Graph of results after adding multi-dimensional dynamic feature matching.

*5.2. Optimization Results Based on Temporal Logic Matching Method*

After the space–time logic matching, the overall MOTA improved to 0.860, while the IDF1 increased to 0.948 and MCTA to 0.878, as detailed in Figure 11. Under the global raster semantic map, with spatial logic matching, the detection accuracy of the overlapping field of view area is improved. The spatial position association between the body and head detection frames was introduced so that when the body detection frame was lost due to the target's intersection, occlusion, or scale change, the head detection could still maintain the continuous tracking and ID of the target.

As shown in Figure 14a, before the addition of logical matching, the interleaved occlusion result for target ID_2 was mislabeled as ID_3. But after adding space–time logical matching, the tracking ID for target ID 2 was kept unchanged even though it was interleaved due to the continuous tracking based on head detection. As shown in Figure 14b, the detection target features at the edges were not significant, causing ID recognition errors, while the addition of logical matching resulted in the correct target matching across cameras due to spatial clustering.

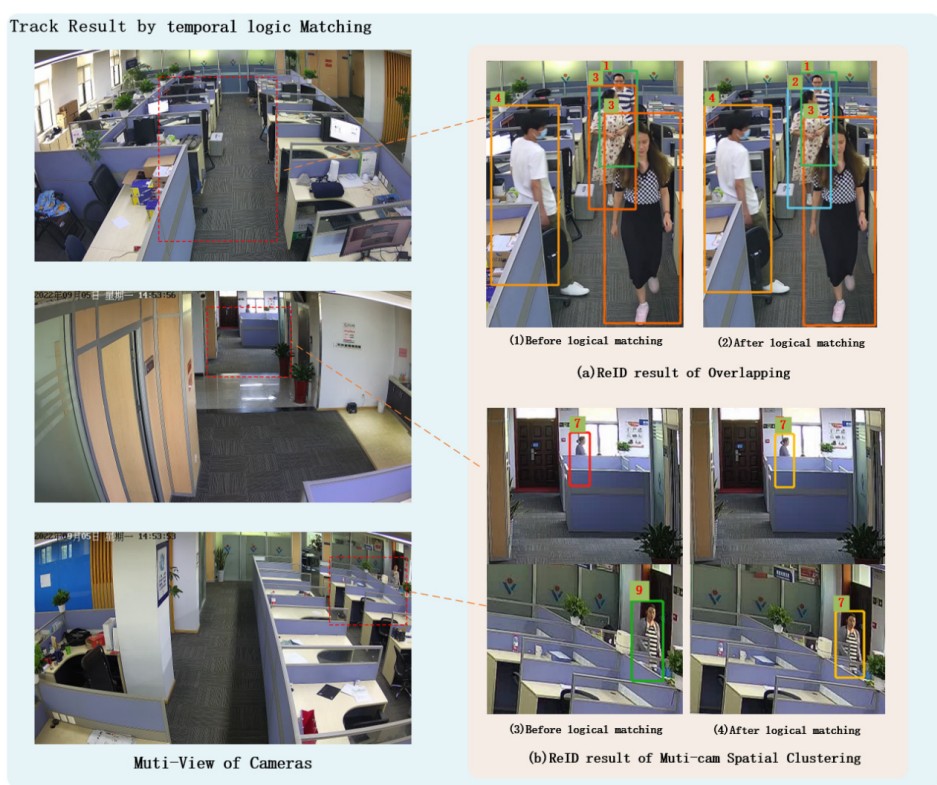

**Figure 14.** Graph of results after adding temporal logic matching.

*5.3. Optimization Results Based on Retrospective Mechanism*

The retrospective mechanism optimized the overall accuracy by using the continuous image sequence information in verifying and correcting some wrong and missed detections in the historical data. The experiment results show that the proposed method improved the overall MOTA to 0.863, IDF1 to 0.981 and MCTA to 0.909.

As shown in Figure 15, Figure 15a shows that the head-based ReID detection result was successfully obtained after the retrospective mechanism. Figure 15b shows the results of sequential image processing tracking of consecutive frames before the retrospective mechanism, which shows that no ReID result was obtained for the head-based detection of target ID_2 at moments t0 and t1; Figure 15c shows the results of consecutive frames after the retrospective processing, which shows that the ReID was completed for target ID_ 2 at moments t0 and t1.

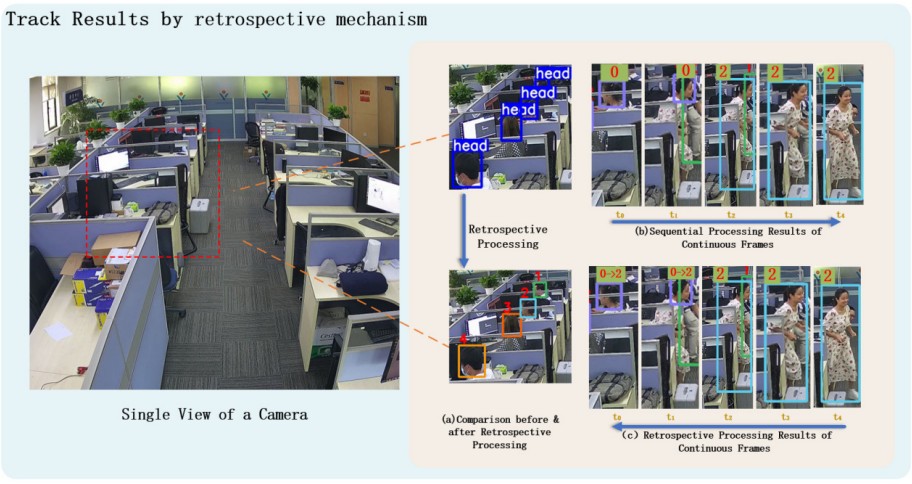

**Figure 15.** Optimization of head tracking results with the addition of the retrospective mechanism.

## 6. Conclusions

An end-to-end multi-target tracking approach was developed based on the transformer's self-attention improvements. In the decoder, the cross-camera target re-recognition results for space–time logic matching and multidimensional feature matching were fused, while a space–time convergence network (STCN) was constructed to pass the re-recognition parameters. The optimizations of the decoder resulted in a 5.0% improvement on MOTA and 2.1% improvement on MCTA. The retrospective mechanism was used to optimize the overall accuracy, which resulted in a 3.5% improvement in IDF1. In the experiments using public datasets, the proposed approach performed well against other algorithms for single-camera tracking. For the MOT01 dataset, the approach method achieved a MOTA value of 0.863 and an IDF1 value of 0.981, significantly improving the overall accuracy of tracking.

This study improved the accuracy and reliability of multi-target tracking under overlapping and non-overlapping FOVs scenarios. However, the current algorithm experimental environment is a circumferential scene with five cameras access and the deployment environment is a server with quad-RTX 2080 SUPER graphics cards. In a larger-scale camera access environment, the algorithm deployment environment and computational efficiency need to be further verified.

## 7. Recommendations and Future Work

The construction process for the raster semantic map in this method remains tedious. In the future, intelligent VR terminals can be combined with related algorithms, such as neural radiation field [42] (NeRF), to achieve more rapid construction of the semantic raster map. Other localization sources (e.g., audio [43], UWB [44], Bluetooth [45]) can also be introduced to assist in target tracking to improve reliability and robustness.

**Author Contributions:** Conceptualization, Y.H. and D.L.; Methodology, Y.H.; Software, S.L. and Y.Y.; Formal analysis, X.C.; Investigation, S.L. and Y.Y.; Data curation, X.C.; Writing—original draft, Y.H., S.L. and Y.Y.; Writing—review & editing, Y.H. and M.W.; Visualization, X.C. and Y.Y.; Supervision, D.L.; Project administration, Y.H.; Funding acquisition, Y.H., D.L. and M.W. All authors have read and agreed to the published version of the manuscript.

**Funding:** This work was supported by the Key Research & Development of Hubei Province (2020BIB006); The Natural Science Foundation of Hubei Province(2020CFA001); The Key Research & Development of Hubei Province (2020AAA004).

**Data Availability Statement:** The data that supports the findings of this study are available from the corresponding author upon reasonable request.

**Acknowledgments:** The authors are sincerely grateful to the editors as well as the anonymous reviewers for their valuable suggestions and comments that helped us improve this paper significantly.

**Conflicts of Interest:** The authors declare no conflict of interest.

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
