# Peer review of "An Improved End-to-End Multi-Target Tracking Method Based on Transformer Self-Attention"

_remotesensing, doi:10.3390/rs14246354_

Round 1

Reviewer 1 Report

This paper introduces an improved multi-target tracking method and the results show that this method can effectively improve accuracy. However, there are too many concepts in this paper that may make the readers confused. I suggest the authors reduce some content to make their points more straightforward.

Minor comments

1. The abstract should not include references.

2. Some references should be added to support the view proposed from line 41 to 46.

3. Line 49 follow instead of follow.

4. Please explain Figure 1, for example, what are A and B represent? What is the full name of IoU. The same is in Figure 2. Or if these pictures are well-known to the readers, they are unnecessary.

5. Figure 3 can be improved. For example, the three subplots in Semantic discrimination should be integrated into one.

Major comments

1. I don’t think the authors present their work in a proper way. Some aspects are mentioned but don’t show enough innovation, such as the normalization of complex scenes. It is a little strange that only a small part of Section 2.2.3.3 introduces the normalization of complex scenes.

2. The authors don’t need to give titles in the introduction part. Please check the common format used in this journal. Besides, this paper is too long. As far as I know, some part is not necessary. For example, in the Research Background, some related methods are introduced. If TBA is better than TBD in MTMCT, then it is not necessary to introduce TBD, or just one or two sentences are enough.

3.. What does end-to-end mean? Because it is not mentioned in Methods. The problem is that there is too much the authors want to show. Please simplify the context, if there are many innovations, the authors could split them into different papers.

4. The structures of this paper need to be improved. For example, it is not a whole processing chain that is proposed, only some modules or algorithms. Therefore, the authors should specify these improvements, instead of introducing the general procedure and the proposed procedure and letting the readers find out the difference.

Reviewer 2 Report

The work is quite interesting. The authors have done a lot of good work. However, there are some comments and suggestions.

  1. Abstract should be rewritten in a more general form, highlighting what new data was obtained, how it broadened our knowledge, and whether it made a new contribution to the methodology of studying.
  2. A few acronyms that are used throughout the manuscript are not defined. As a result, it's advised to define each acronym when it appears for the first time. There are also a number of grammar and punctuation errors that need to be fixed.
  3. The introduction is not enough. In the part of the introduction, it is necessary to explain clearly the prerequisites obtained by previous scholars. I would like to see a restructure of the introduction: the author should update the references list with recent publications such as https://doi.org/10.3390/rs14051149. The contribution in the Introduction section should be rearranged. the proposed innovation is not reflected.
  4. It is a scientific research paper so it is suggested to delete the “1.1” from line No 35. Continued directly from “1. Introduction/ Research Background”. From the related work, all figures will move to the methodology section or any other suitable place. Related work always highlighted what other research scholars have achieved so, therefore, no need to present any of your methodology figures in this section. And “1.3 problems” should arrange at the end of the introduction section without a subsection.
  5. The paper’s logical explanation is poorly written. To explain the algorithm's mechanism in a logical way, the method section should be revised. Figures 3, 4, 5, 6, and 7 needed some more logical explanations to support the figure.
  6. As the object detection accuracy assessment index appears extremely desirable, it would have been more abundant if it has been shown in graphs in addition to tables. Similarly, more test data should be used to enhance the method's soundness.
  7. The conclusion might be strengthened and rewritten to place more emphasis on the study's findings than on any problems or implications. What is innovative about the research? What are the "limitations of the research"? Please put the section "Recommendations and Future Work" at the end.

Reviewer 3 Report

The manuscript presents an end-to-end multi-target tracking method. However, there are some important problems that need to be clarified. See attached file.

Round 2

Reviewer 1 Report

The authors have answered my questions comprehensively. In my opinion, the paper can be accepted in its present form.

Reviewer 3 Report

My concerns have been addressed.